# Chinese Cabbage Changes Its Release of Volatiles to Defend against *Spodoptera litura*

**DOI:** 10.3390/insects13010073

**Published:** 2022-01-10

**Authors:** Yuan-Wen Du, Xiao-Bin Shi, Lin-Chao Zhao, Ge-Ge Yuan, Wei-Wei Zhao, Guo-Hua Huang, Gong Chen

**Affiliations:** 1Hunan Provincial Key Laboratory for Biology and Control of Plant Diseases and Insect Pests, Hunan Agricultural University, Changsha 410128, China; duyuanwen112@163.com (Y.-W.D.); zhaolinchao946@163.com (L.-C.Z.); gegeyuan206@163.com (G.-G.Y.); 2College of Plant Protection, Hunan Agricultural University, Changsha 410128, China; 3Hunan Plant Protection Institute, Hunan Academy of Agricultural Sciences, Changsha 410125, China; shixiaobin@hunaas.cn; 4Plant Protection and Quarantine Institution, Shimen County Agriculture and Rural Bureau, Changde 415399, China; smxzbzj@126.com

**Keywords:** Chinese cabbage, plant-defense responses, *Spodoptera litura*, *Microplitis similis*

## Abstract

**Simple Summary:**

Biological control is an important direction for pest control in the future, and chemical ecology is an indispensable part of biological control. Therefore, we tested the selection of *Spodoptera litura* and parasitic wasps on the volatiles of different treatments of cabbage and collected and analyzed the volatiles of different treatments of cabbage. This study found that cabbage was fed by *Spodoptera litura* to produce volatiles to avoid *Spodoptera litura* while also attracting *Microplitis similis.* As a result, some compounds were found to be related to the behavior of *Spodoptera litura* and *Microplitis similis*. These results provide a theoretical basis for searching for biological control resources and chemical control.

**Abstract:**

Plants respond to herbivorous insect attacks by releasing volatiles that directly harm the herbivore or that indirectly harm the herbivore by attracting its natural enemies. Although the larvae of *Spodoptera litura* (the tobacco cutworm) are known to induce the release of host plant volatiles, the effects of such volatiles on host location by *S. litura* and by the parasitoid *Microplitis similis*, a natural enemy of *S. litura* larvae, are poorly understood. Here, we found that both the regurgitate of *S. litura* larvae and *S. litura*-infested cabbage leaves attracted *M. similis*. *S. litura* had a reduced preference for cabbage plants that had been infested with *S. litura* for 24 or 48 h. *M. similis* selection of plants was positively correlated with the release of limonene; linalool and hexadecane, and was negatively correlated with the release of (*E*)-2-hexenal and 1-Butene, 4-isothiocyanato. *S. litura* selection of plants was positively correlated with the release of (*E*)-2-hexenal, 1-Butene, 4-isothiocyanato, and decanal, and was negatively correlated with the release of limonene, nonanal, hexadecane, heptadecane, and octadecane. Our results indicate that host plant volatiles can regulate the behavior of *S. litura* and *M. similis*.

## 1. Introduction

Plants that are not infested with herbivorous insects emit only trace quantities of volatile organic compounds (VOCs). When infested by herbivores, in contrast, some plants emit large quantities of volatiles (herbivory-induced plant volatiles: HIPVs) [1], including terpenoids, green leaf volatiles, nitrogen-containing compounds, nitriles, oximes, aldehydes, alcohols, ketones, esters, ethers, and carboxylic acids [2,3]. These volatiles can be used to communicate with neighboring plants and insects, and they are also the precursors of plant defense measures.

Over a long period of coevolution, plants and insects have developed strategies to avoid each other’s defense systems [4,5]. In response to attack by herbivorous insects, plants activate inducible defenses. Many of these inducible defenses directly target the herbivore’s physiology and may protect the plant against further damage [6]. The induced insect resistance of plants is usually defined as direct or indirect according to the mode of action [7,8,9]. Direct modes of action involve the production of chemicals by the plant that directly harms the herbivore, whereas the indirect modes of action increase herbivore mortality through the recruitment of natural enemies [10,11,12]. Natural enemies of herbivores use HIPVs to locate their prey [13,14,15,16].

When acting as direct defenses, HIPVs can act as repellants or can reduce larval growth [17]. For example, maize leaves initially infested by *Spodoptera exigua* larvae can repel *S. exigua* and thereby reduce further damage [18]. When laying eggs, *Spodoptera frugiperda* females avoid tobacco plants that have been harmed by *S. frugiperda* [19]. HIPVs, however, can have different effects on different insects. For example, *S*. *frugiperda* larvae prefer to feed on maize leaves that have been infested by *S*. *frugiperda* larvae [20].

Regarding indirect defenses involving HIPVs, two natural enemies of the thrips *Thrips tabaci*, *Amblyseius cucumeris*, and *Orius similis* use volatiles released by thrips-damaged hosts to locate their prey [21]. The parasitoid *Aphidius gifuensis*, a natural enemy of *Sitobion avenae*, is more attracted to *S. avenae*-infested wheat than to non-infested wheat [22]. Similarly, the parasitoid *Cardiochiles nigriceps*, a natural enemy of *Heliothis virescens* but not of *Helicoverpa zea*, uses plant volatiles to distinguish between *H. virescens*- and *H. zea*-infested plants [23]. Furthermore, intermittent exposure to the volatiles emitted from artificially damaged *Arabidopsis* has been shown to induce defensive responses in undamaged neighboring plants [24].

The current research concerns the effects of HIPVs on *S. litura* and *M**. similis*. *S. litura* is a serious pest, i.e., its larvae consume the leaves of tobacco and many other valuable crop plants. Although HIPVs are known to increase the attractiveness of sex pheromone lures used to trap *S. litura* adults, little is known about the effects of HIPVs on the feeding behavior of *S. litura* larvae [25]. The larvae of *S. litura* are hosts for the parasitoid wasp *M. similis*, which is an important biological control agent of the herbivore. *M. similis* can reduce the weight gain and feed intake of the *S. litura* larvae and can increase larval mortality [26,27]. In addition, *M. similis* is a vector of insect ascoviruses-vesicle virus. The “bee-venom” coordinated control mode can expand the spread radius of the virus in the host population, thereby greatly improving the spread of *M. similis* biological control effect [28]. In the current research, we assessed the responses of *S. litura* and *M. similis* to HIPVs released by Chinese cabbage leaves or cabbage plants that had been treated in one of six ways, i.e., they were healthy, mechanically damaged, coated with the regurgitate of *S. litura* larvae or infested with *S. litura* larvae for 12, 24, or 48 h.

## 2. Materials and Methods

### 2.1. Plant Material

Chinese cabbage plants (Ju Hong Xin) were grown in a greenhouse at 20 ± 5 °C and with a 12 h:12 h light:dark photoperiod at the Hunan Provincial Key Laboratory for Biology and Control of Plant Diseases and Insect Pests. Experiments included six kinds of plants (all with six fully expanded leaves) or leaves from such plants: undamaged healthy leaves (HP), mechanically damaged plants (MDP: obtained by scratching the leaves of healthy plants 5 times with an insect needle), plants that were dipped in a 5% solution of *S. litura* “regurgitate” (RTP: see next section) for 12 h [29], and plants that were infested with second- to third-instar larvae of *S. litura* (SIP: 1 larva per leaf) for 12 h, 24 h, or 48 h.

### 2.2. Insect

Specimens of the cotton leafworm *S. litura* were collected from the experimental field of Hunan Agricultural University; *M.*
*similis* was collected in a cotton field near Hunan Agricultural University, Changsha, Hunan, China. *S. litura* eggs and *M. similis* were obtained from our laboratory. *M. similis* were reared from *S. litura* larvae as the host. They were reared in a chamber at 27 ± 2 °C, RH = 70 ± 10%, and a 14 h: 10 h L: D photoperiod. Regurgitate from third- to fourth-instar larvae of *S. litura* were collected by gently squeezing the larvae manually and placing a pipette tip on their mouthparts. The regurgitated material was then transferred to Eppendorf tubes and stored at −80 °C.

### 2.3. Olfactory Preferences of M. similis (M. similis Olfactometer Experiments 1–4)

We used a 4-arm olfactometer [30] with one glass sample bottle per arm to assess the olfactory preferences of 3-day-old mated females of *M. similis*. Each of the first three experiments included two empty sample bottles (blanks) and one bottle containing an undamaged, healthy plant. The fourth bottle in experiments 1–3 contained a mechanically damaged plant, a plant treated with larval regurgitate, or a plant infested with *S. litura* for 12 h, respectively. In experiment 4, the sample bottles contained a healthy plant, a mechanically damaged plant, a plant treated with larval regurgitate, or a plant infested with *S. litura* for 12 h. Each assay with four sample bottles was replicated 30 times.

The trap ball was attached above the four selection arms; for each *M. similis* individual, the selection was indicated by its climbing into the trap ball. An LED tube was hung 18.0 cm above an olfactometer, which was placed under a white canvas cover to diffuse the light source from interfering with the selection behavior of the wasps. In the experiment, a medical air compressor with low noise and relatively pure gas was selected as the gas delivery device, and a three-stage oil-water separator was used to filter out impurities in the gas, and then a clean and moist gas was produced after passing it into the filter bottle. By controlling the air compressor, the total gas flow rate is controlled at about 2.5 L min^−1^, and the rotor flow device is adjusted to control the airflow rate through each gas source bottle at about 500 mL min^−1^. Individual airflow was connected to each odor source and converged to a central glass piece where 4 *M. similis* females were released. After thirty minutes following release, the location of each of the wasps was noted. Selection for an odor source was concluded when a wasp was present in one of the four ball traps. The experiment was replicated thirty times with 4 female wasps each time. Between replicates, glassware was cleaned by sequential rinsing in water, acetone, and pentane, after which the glassware was placed in an oven at 250 °C for 3 h.

### 2.4. Behavioral Responses of S. litura Larvae to Chinese Cabbage and Volatile Compounds (S. litura Petri Dish Experiments 1 and 2, and S. litura Olfactometer Experiments 1 and 2)

The preferences of *S. litura* larvae for leaves treated in the indicated ways were compared in Petri dishes (20 cm diameter) containing a piece of filter paper with a drawn line that separated the dish into two equal sides. The experiments were conducted at 28 ± 1 °C and in the dark. In Petri dish experiment 1, the following treatments were compared: CK (blank treatment without plants) vs. healthy leaves, and CK vs. leaves that had been infested by *S. litura* for 12, 24 h, or 48 h. In Petri dish experiment 2, the following treatments were compared: healthy leaves vs. leaves that had been infested by *S. litura* for 12, 24 h, or 48 h (the infesting larvae were removed from the leaves immediately before the experiment began). In each replicate of each treatment of the Petri dish experiments, one second- to third-instar larvae of *S. litura* that had been starved for 24 h was placed in the center of the dish, and its feeding selection was recorded at 15 min. After a larva made a selection, it was removed, and the original leaf was immediately replaced with a new leaf for the next larva to choose. The Petri dish and filter paper were replaced every 3 larvae. The Petri dish experiments were conducted 3 times, each time with 30 replicates per treatment.

A glass Y-tube olfactometer, consisting of a 12 cm long base tube and two 16 cm long arms at a 60° angle with an inner diameter of 2.0 cm, was used to observe the olfactory response of *S. litura* larvae to the volatiles of cabbage leaves. Before the start of each assay, we treated the system with compressed air for 10 min to ensure that the odors in the arms were equivalent. By controlling the air compressor and rotameter, we controlled the gas flow rate at 300 mL min^−1^ and leading to the two arms of the olfactory instrument. Under dark conditions and at 28 ± 1 °C, we released one third-instar larva of *S. litura* at the base of the straight tube. At 20 min from the time of release, if the *S. litura* larva moved half the length of either arm, it was considered to have made a choice. If the larva had not moved half the length of an arm within 20 min, it was recorded as “no choice”. Plants were changed, and glassware was cleaned every five insects made a choice. The treatments and replicate are the same as the Petri dish experiment.

### 2.5. Collection and Analysis of Volatiles via Solid Phase Microextraction

The commercial fiber DVB/CAR/PDMS 50/30 μm (SAAB-57328U, Supelco, Inc., Bellefonte, PA, USA) was mounted in an SPME manual holder (SAAB-57330U, Supelco, Inc., Bellefonte, PA, USA). For each analysis, 5 g of sample was placed in a 40-mL vial and was spiked with 0.02 mL of an internal standard (11.03 g/L heneicosane). After each vial was placed at 30 °C to equilibrate, the septum was pierced with the SPME needle. Fibers were exposed to the sample headspace for 60 min. After the extraction time, fibers were retracted into the needle, immediately transferred to the injection port of a GC, and desorbed at 230 °C for 1 min.

VOCs were analyzed with a GC connected to a mass spectrometer (MS) (Agilent 7890A-5975C). Separation was carried out on a Supelcowax 10 fused silica capillary column 30 m × 0.25 mm ID × 0.25 μm film thickness (Agilent J&W DB-5MS). On the DB-5 ms column, the initial oven temperature was kept at 60 °C for 2 min and was then increased to 100 °C at a programmed rate of 3 °C min^−1^ and then to 190 °C at a rate of 2 °C min^−1^. The inlet was operated in splitless injection mode, and the injector temperature was maintained at 220 °C with a constant flow rate of 2.0 mL min^−1^. The mass spectra were scanned at 70 eV over a mass range from *m*/*z* 40 to 450. The quantities of the major components of the blends were estimated based on the peak areas of the compounds compared to the peak areas of the internal standards. Compounds were identified by comparing the spectra obtained from the samples with those from a reference database (NIST mass spectral library).

### 2.6. Statistical Analysis

Before analysis, data were checked for normality and homogeneity of variance; data were log-transformed as needed. The raw data were sorted with Microsoft Excel 2016 and were then analyzed with SPSS 22.0 software (IBM SPSS Statistics 22.0). The data concerning the effects of treatments on the selection of *S. litura* were compared with the Chi-square test (*p* < 0.05). Data of *M. similis* experiments 1–4 and the release amount from leaves of different treatments were carried out by one-way ANOVA with post-hoc contrasts by Tukey’s honestly significant difference test (α = 0.05). The relationship between volatile release and host searching behavior of *M. similis* and *S. litura* was analyzed by SPSS software, and the Pearson correlation coefficient was used to analyze the correlation after the data basically conformed to linear correlation, normal distribution, and no abnormal value. The figures were plotted using GraphPad Prism 8 (GraphPad Software, San Diego, CA, USA).

## 3. Results

### 3.1. Olfactory Preferences of M. similis (M. similis Olfactometer Experiments 1–4)

The attraction of *M. similis* significantly differed among treatments in the four experiments. In the first experiment, the attraction of *M. similis* did not differ for healthy and mechanically damaged plants, but both were more attractive than the blank odor source (*p* < 0.05; Figure 1a). In the second experiment, plants treated by *S. litura* regurgitate were much more attractive than healthy plants or the blank odor source (*p* < 0.05; Figure 1b). In the third experiment, plants infested by *S. litura* larvae were much more attractive than healthy plants or the blank odor source (*p* < 0.05; Figure 1c). In the last experiment, plants infested by *S. litura* larvae were the most attractive, mechanically damaged plants, and the blanks odor source were the least attractive, and plants treated with *S. litura* regurgitate were intermediate in their attractiveness (*p* < 0.05; Figure 1d).

### 3.2. Feeding and Olfactory Selecting Behavior of S. litura Larvae (S. litura Petri Dish Experiment 1 and 2, and S. litura Olfactometer Experiments 1 and 2)

In the first Petri dish experiment, *S. litura* preferred healthy leaves over no leaves (blank) and preferred *S. litura*-infested leaves over no leaves (blank) (*p* < 0.001; Figure 2a). In the second Petri dish experiment, *S. litura* larvae showed no significant preference for healthy leaves vs. leaves that had been infested with *S. litura* for 12 h; as the infestation time increased to 24 and 48 h; however, *S. litura* clearly preferred healthy leaves over *S. litura*-infested leaves (Figure 2b). In “Y” type olfactometer experiments, the selection rate of healthy plants and of *S. litura*-infested plants was significantly higher than that of no plant, but an increase in the infestation time tended to decrease the selection of infested plants (*p* < 0.01; Figure 2c), *S. litura* larvae showed no significant preference between healthy plants and plants that had been infested with *S. litura* for 12 h; however, *S. litura* larvae clearly preferred healthy plants over plants that had been infested with *S. litura* for 24 and 48 h (*p* < 0.001; Figure 2d).

### 3.3. Quantities of Total and Six Classes of Volatiles Released

The quantity of volatiles collected and identified was highest for plants infested with *S. litura* for 48 h was lowest for undamaged plants, mechanically damaged plants, and regurgitate-treated plants; and was intermediate for plants infested with *S. litura* for 12 h and 24 h (*p* < 0.05; Figure 3).

The volatiles were assigned to the following six main classes of HIPVs in Brassicaceae [31]: green leaf volatiles, ester, aldehydes, terpenoids, ketones, and alkanes (Figure 3). The release of esters, aldehydes, and ketones by Chinese cabbage did not significantly differ between healthy plants vs. mechanically damaged plants or plants infested with *S. litura* for 12 h. There was no significant difference in the release of any class of volatiles between healthy and mechanically damaged plants. The regurgitate treatment increased the release of alkanes but not of other volatiles. The emission of terpenoids, ketones, and alkanes tended to increase, but the emission of green leaf volatiles decreased with the increase in *S. litura* infestation time.

### 3.4. Quantities of 16 Specific Compounds Released

GC-MS analysis of the headspace volatiles produced by each treatment of Chinese cabbage detected 17 compounds (Appendix A). Geranyl acetone has been compared with other categories mentioned above, and we have compared the release of the remaining 16 compounds. (Figure 4): (*E*)-2-hexenal; (*Z*)-3-henxen-1-ol; butane,1-isothiocyanato; 1-Butene,4-isothiocyanato; (*Z*)-3-hexenyl acetate; limonene; ally isothiocyanate; linalool; nonanal; decanal benzyl isothiocyanate; tetradecane; pentadecane; hexadecane; heptadecane; octadecane. Butane,1-isothiocyanato was not detected in the *S. litura* regurgitate treatment, and limonene was only detected from S. *litura*-infested plants. The release of nonanal was increased by *S. litura* infestation for 24 and 48 h and tended to be increased by mechanical damage, regurgitate, and *S. litura* infestation for 12 h. Regurgitate increased the release of nonanal, pentadecane, and heptadecane but reduced the release of butane,1-isothiocyanato, 1-Butane,4-isothiocyanato, and ally isothiocyanate. *S. litura* infestation increased the release of pentadecane and heptadecane, but the release of 1-Butane,4-isothiocyanato and decanal. The effect of *S. litura* infestation varied with infestation time. *S. litura* infestation for 12 h increased the release of (*Z*)-3-hexen-1-ol, butane,1-isothiocyanato, (*Z*)-3-hexenylacctate, limonene; allyisothiocyanate, linalool, nonanal, and henxadecane, the release of these compounds was decreased; however, by *S. litura* infestation for 24 h. The quantity of (*E*)-2-hexenal released was lower for plants infested with *S. litura* for 12 h than for healthy plants, and (*E*)-2-hexenal was not detected from plants infested with *S. litura* infested for 24 and 48 h. Butane,1-isothiocyanato was also not detected from plants infested with *S. litura* for 24 or 48 h.

### 3.5. Relationships between the Release of Classes of Volatiles and the Host Searching Behavior of M. similis and S. litura

Linear regression was used to analyze the linear relationships between the quantities of classes of volatile emissions and the selection rate of *M. similis* and *S. litura*. According to linear regression, the *M. similis* selection rate was positively related to the release of terpenoids (*p* < 0.05, Table 1).

According to linear regression, the selection rate of *S. litura* was positively related to the release of esters but was negatively related to the release of aldehydes and ketones (*p* < 0.05, Table 1).

### 3.6. Relationship between the Release of Specific Volatiles by Chinese Cabbage and the Host Searching Behavior of M. similis and S. litura

According to linear regression, the *M. similis* selection rate was positively related with the release of limonene, linalool, and hexadecane but was negatively related with the release of (*E*)-2-hexenal and 1-Butene, 4-isothiocyanato (*p* < 0.05, Table 2).

According to linear regression, the selection rate of *S. litura* was positively related with the release of (*E*)-2-hexenal, 1-Butene, 4-isothiocyanato, and decanal but was negatively related with the release of limonene, nonanal, hexadecane, heptadecane, and octadecane (*p* < 0.05, Table 2).

## 4. Discussion

The tritrophic interactions among plants, herbivores, and their parasitoids and influence each other in the long course of coevolution [12]. Plants use an array of biochemical and morphological properties to defend against attacks by herbivores. For direct defense, plants produce organic compounds that interfere with the behavior of herbivores [10,32,33]. For indirect defense, the HIPVs produced by plants can attract natural enemies of herbivores [6,9,23,34]. This study shows that Chinese cabbage releases volatiles that can suppress *S. litura* larvae from choosing leaves that are already exploited. However, when the larvae choose to feed on unattacked leaves, they face the risk of being found by *M. similis.* These patterns support the proposition that plants have evolved to manipulate the blend of volatiles that are emitted in order to maximize both direct and indirect defenses against herbivory.

Olfactory receptors can be used by herbivorous insects to help locate hosts or to avoid danger [35,36,37,38]. Studies on the oviposition behavior of *S. litura* have shown that the adults of *S. litura* can rely on plant volatiles to find the most suitable hosts but to avoid plants already infested by *S. litura* larvae [39,40]. Before the current study, however, little was known about how host plant volatiles affects the feeding of *S. litura* larvae. We found that *S. litura* larvae prefer undamaged to damaged cabbage tissue and showed significant responses to HIPVs (Figure 2b,d). The characteristics of this feeding behavior are consistent with those of other Noctuidae insects. *S. exigua*, for example, showed obvious antifeedant and evasive responses to corn HIPVs [17].

In the current study, we found that *M. similis* females were attracted to both *S. litura* regurgitate-treated plants and *S. litura*-infested plants, but when given a choice among volatiles produced by healthy, mechanically damaged, regurgitate-treated, and *S. litura*-infested plants, they strongly preferred volatiles produced by *S. litura*-infested plants (Figure 1). In response to damage caused by herbivorous insects, plants can regulate lipoxygenase, isoprenoid, and shikimate pathways in order to release compounds that attract natural enemies of herbivores for indirect defense [41]. This phenomenon has been previously reported. When infested by *S. exigua*, for example, *Zea mays*, *Nicotiana tabacum*, and *Gossypium hirsutum* release volatiles that attract wasps that parasitize *S. exigua* [13,23,42].

To determine whether a single factor in *S. litura* larval feeding affects the behavior of parasitic wasps, we simulated the effects of mechanical damage and herbivore regurgitate on the behavior of *M. similis*. We found that, compared with responses to clean air, *M. similis* females showed strong responses to the volatiles of regurgitate-treated plants but not to the volatiles of mechanically damaged plants (Figure 1a,b). Mechanical damage to plants also failed to attract *Microplitis palidipes* [43]. When comparing the effects of the treatments on the release of classes of volatiles, we found that there was no significant difference between the mechanical damage treatment and the healthy treatment, but that the *S. litura* regurgitate treatment reduced the release of aldehydes and alkanes (Figure 3). We found that the *S. litura* regurgitate treatment reduced the release of butane,1-isothiocyanato and 1-Butene,4-isothiocyanato, and increased the release of nonanal, pentadecane, and heptadecane (Figure 4). A reduction in the release of specific volatiles by plants has also been found for other Noctuidae. For example, glucose oxidase in *Helicoverpa zea* regurgitate can reduce the release of nicotine in tobacco [44]. In another study, the regurgitate of *S. frugiperda* reduced the release of corn HIPVs but did not affect the attraction of the parasitoid *Cotesia marginiventris* to *S. frugiperda*-infested corn [45]. As part of their long-term coevolution with host plants, herbivores may reduce the release of HIPVs so as to reduce the indirect defense involving the attraction of natural enemies.

To simulate the feeding of *S. litura* in the current study, we simply soaked the plants in regurgitate. The total amount of volatiles released was smaller after regurgitate treatment than after larval infestation (Figure 3), which is reasonable because *S. litura* larvae will both produce regurgitate and destroy a substantial quantity of host plant tissue. According to the results discussed above, we speculate that the backflow of *S. litura* may have elicitors to change the volatiles of Chinese cabbage, and the induced volatiles will affect the selection behavior of *M. similis*.

Previous studies have shown that the length of time that plants are infested by herbivorous insects will affect the diversity and richness of HIPVs [1,46,47]. For example, tomato plants released more (3*Z*)-3-hexen-1-yl acetate, *β*-ocirene, and *β*-caryophyllene after 12 h than after 6 h of *Trichoplusia ni* infestation [48]. In this study, we found that the effects of HIPVs released from *S. litura*-infested Chinese cabbage on the selection behavior of *S. litura* and *M. similis* differed depending on the duration of the infestation. When the plants were infested by *S. litura* larvae for 12 h, the released volatiles significantly affected host location by *M. similis* but not by *S. litura* (Figure 1c and Figure 2d). After the plants were infested by *S. litura* for 12 h, the release of (*Z*)-3-hexen-1-ol, (*Z*)-3-hexenyl acetate, ally isothiocyanate, linalool, and octadecane increased relative to levels in healthy plants; limonene was detected only with infested plants, and butane,1-isothiocyanato only appeared after infestation for 12 h. However, *S. litura* showed a reduced preference for plants that were infested by *S. litura* larvae for 24 h (Figure 2d). The release of (*Z*)-3-hexen-1-ol, butane, 1-isothiocyanato, (*Z*)-3-hexenyl acetate, and linalool was highest after 12 h of *S. litura* infestation and then gradually decreased. Release of 2-hexenal, (*Z*)-3-hexen-1-ol, (*Z*)-3-hexenyl acetate, and linalool was higher early during *S. litura* infestation than later during *S. litura* infestation. The release of nonanal, pentadecane, heptadecane and octadecane increased with the infesting time of *S. litura*. Chinese cabbages were able to reduce their attractiveness to *S. litura* larvae after they had been infested for 24 h but not for 12 h. This is because plants require time for their resistance genes to recognize the effectors of herbivorous insects and to induce immune responses [49]. Plants differ in their immune response times, and Chinese cabbage in our experiment required 24 h to produce an immune response.

Correlation analysis indicated that the selection rates of *M. similis* and *S. litura* were affected by volatile class and identity. The results showed that the increased release of limonene, linalool, and hexadecane increased the attractiveness of plant tissue to *M. similis* but that the increased release of (*E*)-2-hexenal and 1-Butene, 4-isothiocyanato reduced the attractiveness of plant tissue to *M. similis*. The increased release of (*E*)-2-hexenal, 1-Butene, 4-isothiocyanato, and decanal increased the attractiveness of plant tissue to *S. litura*, but the increased release of limonene, nonanal, hexadecane, heptadecane, and octadecane reduced the attractiveness of plant tissue to *S. litura*. We can find the herbivore and the parasitoid are expressing opposite attraction to some volatiles; it has become clear that plants respond to arthropod herbivory with the induced production of volatiles, but because they have different degrees of evolution in three trophic level interactions (tritrophic interaction), the effects of HIPVs on *S. litura* and *M. similis* would also be different. Plants release specific volatile substances to affect the behavior of different insects to protect themselves. In addition, the difference in recognition and acceptance of the same HIPVs between *S. litura* and *M. similis* is mainly due to their different olfactory recognition system.

Some of the volatiles detected in our research have been found to affect insect behavior in previous studies. For example, linalool elicits a strong olfactory response in the parasitoid *Anagrus nilaparvatae*, and 2-hexenal elicited a strong electroantennogram (EAG) response of meadow moth and *Anomala corpulenta* [50,51]. Glucosinolates are unique sulfur-containing secondary metabolites of cruciferous plants; one of the main decomposition products of Glucosinolates is 1-Butene, 4-isothiocyanato, which plays a role in host location by the herbivorous insects *Plutella xylostella* and *Phyllotreta vittula* [52,53,54]. Decanal, a deca-carbon aldehyde, is attractive to *Phthorimaea operculella* and *Locusta migratoria* [55]. The HIPVs contain a complex mixture of many compounds, the composition of which may vary with herbivore species, herbivore developmental instar, plant tissue, and abiotic conditions [56,57]. Recent research has shown that HIPVs could be utilized in agriculture in different strategies [58]. These compounds can be used singly or in combination to dispense volatile chemicals that affect arthropod foraging behavior; apply inducing agents that alter the attractiveness of plants; breed for or otherwise create crops with enhanced volatile emissions through genetic engineering, thus producing better synergistic effects on parasitoid/herbivore attractions. In order to understand how these compounds affected the behavior of *M. similis* and *S. litura*, we need to determine whether these compounds standard affect the behavior of *M. similis* and *S. litura*. Although our results indicate that HIPVs can attract *M. similis* and repel *S. litura*, the molecular mechanisms underlying herbivore-induced direct and indirect defenses remain to be determined.

## Figures and Tables

**Figure 1 insects-13-00073-f001:**
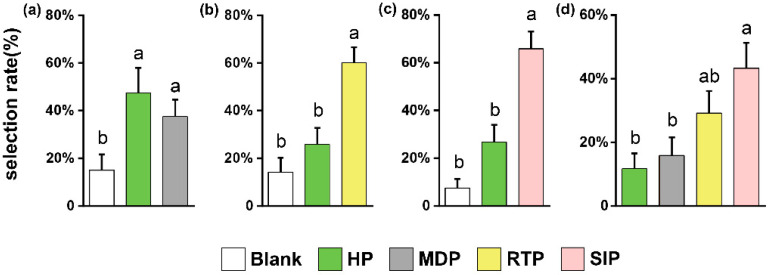
Percentage of *M. similis* females attracted to treatments in a four-arm olfactometer. Preference of *M. similis* for (**a**) Mechanically damaged vs. healthy plants and blanks *(M. similis* olfactometer experiment (1); (**b**) RTP-treated plants vs. healthy plants and blanks (*M. similis* olfactometer experiment (2); (**c**) Plants infested with *S. litura* for 12 h vs. healthy plants and blanks (*M. similis* olfactometer experiment (3), and (**d**) Mechanically damaged, RTP-treated, *S. litura*-infested, and healthy plants (*M. similis* olfactometer experiment (4). Blank: empty odor source (no plants); HP: healthy plants; MDP: mechanically damaged plants; RTP: “regurgitate”-treated plants; SIP: *S. litura*-infested plants. Values are means + SE. Bars labeled with different letters are significantly different by Tukey’s HSD test (*p* < 0.05; one-way ANOVA, α = 0.05).

**Figure 2 insects-13-00073-f002:**
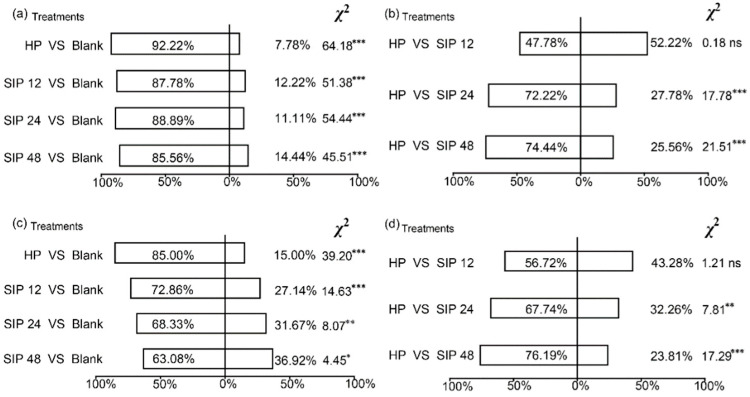
Percentage of *S. litura* attracted to no plants (blank), healthy Chinese cabbage plants (HP), plants infested with *S. litura* (SIP; for 12, 24, or 48 h) in *S. litura* Petri dish experiments 1 (**a**) and 2 (**b**) and in *S. litura* olfactometer experiments 1 (**c**) and 2 (**d**). *, **, ***, and ns indicate *p* < 0.05, <0.01, <0.001, and not significant, respectively, according to a *χ*^2^ test.

**Figure 3 insects-13-00073-f003:**
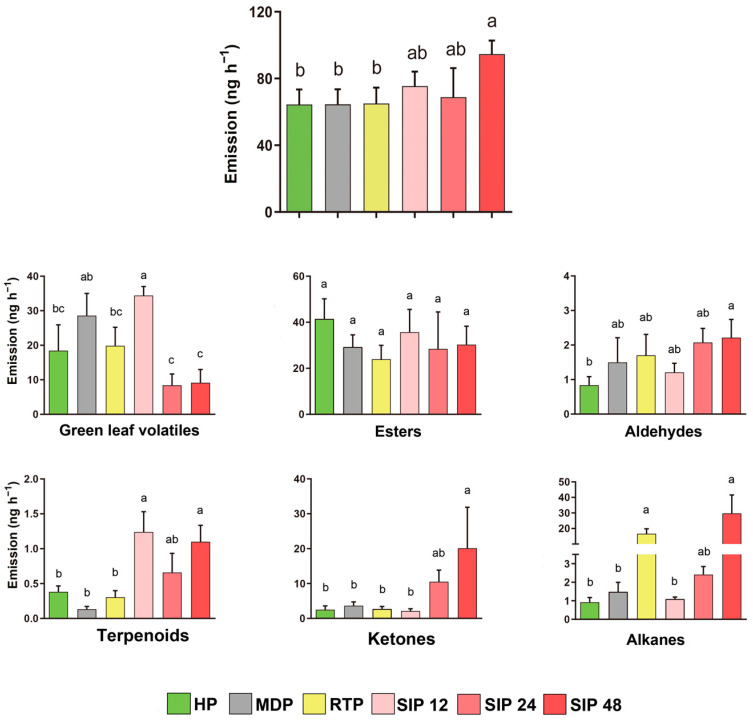
Total quantities of volatile organic compounds (VOCs) released by Chinese cabbage plants in response to six treatments. Total quantities of six classes of volatiles released by Chinese cabbage plants in response to six treatments. HP: healthy plants; MDP: mechanically damaged plants; RTP: regurgitate-treated plants; SIP: plants infested with *S. litura* for 12, 24, or 48 h. Values are means + SE. Bars labeled with different letters are significantly different by Tukey’s HSD test (*p* < 0.05; one-way ANOVA, α = 0.05).

**Figure 4 insects-13-00073-f004:**
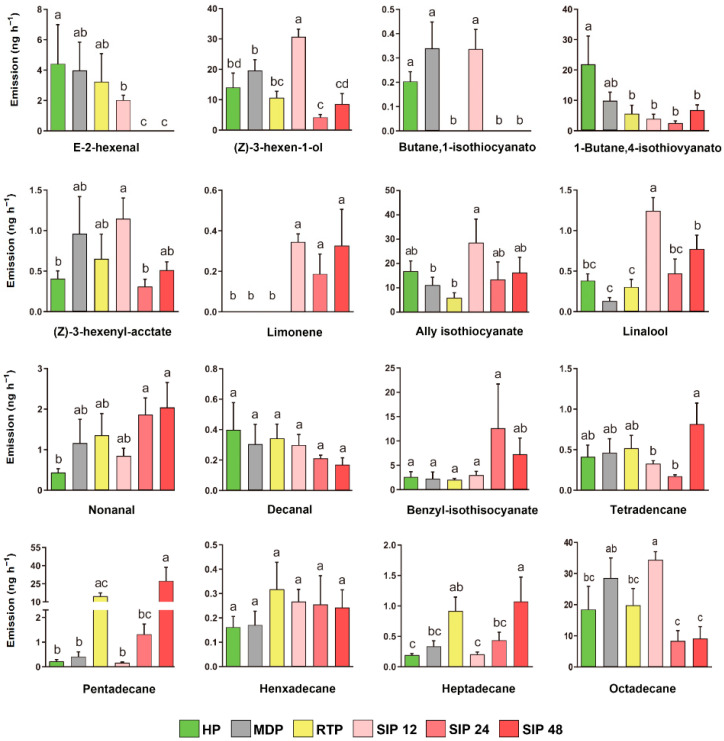
Quantities of specific volatiles released by Chinese cabbage plants in response to six treatments. HP: healthy plants; MDP: mechanically damaged plants; RTP: regurgitate-treated plants; SIP: plants infested with *S. litura* for 12, 24, or 48 h. Values are means + SE. Bars labeled with different letters are significantly different by Tukey’s HSD test (*p* < 0.05; one-way ANOVA, α = 0.05).

**Table 1 insects-13-00073-t001:** Linear relationships between quantities of classes of volatiles emitted by Chinese cabbage plants under different treatments and the selection rates of *M. similis* and *S. litura*.

Class	Linear Regression Equation	*R* ^2^	*p*
Green leaf volatiles	y_1_ = 0.0122x − 0.0576	0.3775	0.6542
	y_2_ = 0.0031x + 0.6681	0.1609	0.5782
Esters	y_1_ = −0.1427x + 0.6413	0.3946	0.5084
	y_2_ = 0.0148x + 0.2220	0.8530	0.0382
Aldehydes	y_1_ = 0.0963x + 0.1238	0.0587	0.1475
	y_2_ = −0.1305x + 0.9298	0.8681	0.0259
Terpenoids	y_1_ = 0.2537x + 0.1194	0.7103	0.0443
	y_2_ = −0.1545x + 0.8538	0.4286	0.3579
Ketones	y_1_ = −0.0035x + 0.3651	0.0326	0.2442
	y_2_ = −0.0091x + 0.8029	0.6722	0.0182
Alkanes	y_1_ = 0.0037x + 0.2313	0.0372	0.4786
	y_2_ = −0.0045x + 0.7619	0.4698	0.3374

Pearson correlation coefficient (*R*
^2^) between different classes of volatiles released and selection rate of *M. similis* and *S. litura*. “x” in the equations represents the number of different classes of volatiles released. “y_1_” represents the selection rate of *M. similis* as affected by treatments. “y_2_” represents the selection rate of *S.*
*litura* as affected by treatments, significantly correlation at *p* < 0.05.

**Table 2 insects-13-00073-t002:** Linear relationships between quantities of specific volatiles emitted by Chinese cabbage plants under different treatments and the selection rates of *M. similis* and *S. litura*.

Number	Compound	Linear Regression Equation	*R* ^2^	*p*
1	*E*-2-hexenal	y_1_ = −0.1582x + 0.7733	0.9729	0.0118
		y_2_ = 0.0471x + 0.6524	0.9286	0.0195
2	(*Z*)-3-hexen-1-ol	y_1_ = 0.0106x + 0.0504	0.3947	0.0931
		y_2_ = 0.0025x + 0.6880	0.0926	0.6650
3	Butane, 1-isothiocyanato	y_1_ = 0.0308x + 0.2432	0.0011	0.4750
		y_2_ = 0.3384x + 0.6775	0.3567	0.3794
4	1-Butene, 4-isothiocyanato	y_1_ = −0.0148x + 0.4032	0.6497	0.0320
		y_2_ = 0.0087x + 0.6470	0.6842	0.0164
5	(*Z*)-3-hexenyl acetate	y_1_ = 0.2949x + 0.0161	0.4216	0.9269
		y_2_ = −0.0011x + 0.7239	0.0003	0.9830
6	Limonene	y_1_ = 0.1476x + 0.1863	0.7317	0.0230
		y_2_ = −0.4678x + 0.8237	0.6379	0.0123
7	Allyl isothiocyanate	y_1_ = 0.0083x + 0.1218	0.2929	0.4685
		y_2_ = 0.0018x + 0.6892	0.0170	0.8381
8	Linalool	y_1_ = 0.2530x + 0.1196	0.7105	0.0132
		y_2_ = −0.0775x + 0.7788	0.1037	0.7103
9	Nonanal	y_1_ = 0.0929x + 0.1616	0.0626	0.1375
		y_2_ = −0.1147x + 0.8738	0.8862	0.0179
10	Decanal	y_1_ = −1.870x + 0.8797	0.3402	0.5150
		y_2_ = 0.9045x + 0.4791	0.9739	0.0083
11	Benzyl isothiocyanate	y_1_ = 0.1529x − 0.1278	0.1784	0.1890
		y_2_ = −0.0126x + 0.8033	0.3965	0.3324
12	Tetradecane	y_1_ = −0.8443x + 0.6144	0.2143	0.9933
		y_2_ = −0.1241x + 0.7769	0.1323	0.6718
13	Pentadecane	y_1_ = 0.0039x + 0.2349	0.0356	0.4825
		y_2_ = −0.0047x + 0.7575	0.4602	0.3447
14	Hexadecane	y_1_ = 1.7680x − 0.1478	0.6716	0.0163
		y_2_ = −1.560x + 1.085	0.6310	0.0205
15	Heptadecane	y_1_ = 0.0553x + 0.2272	0.0161	0.4896
		y_2_ = −0.1963x + 0.8132	0.6293	0.0228
16	Octadecane	y_1_ = 0.4606x + 0.1868	0.0128	0.2183
		y_2_ = −0.9494x + 0.9098	0.5175	0.0263

Pearson correlation coefficient (*R* ^2^) between different classes of volatiles released and selection rate of *M. similis* and *S. litura*. “x” in the equations represents the quantity of specific volatiles released. “y_1_” represents the selection rate of *M. similis* as affected by treatments, and “y_2_” represents the selection rate of *S. litura* as affected by treatments, significantly correlation at *p* < 0.05.

## Data Availability

Data is contained within the article or Appendix A. The data presented in this study are available in [Appendix A].

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
