# Peer review of "Chinese Cabbage Changes Its Release of Volatiles to Defend against *Spodoptera litura"

_insects, 2022, doi:10.3390/insects13010073_

Round 1

Reviewer 1 Report

The authors present a classical study of tri-trophic interactions but is original by including both the herbivore and the natural enemy responses to plant volatiles generated by the herbivore host plant, treated or untreated, and by providing a detailed analysis of the VOCs and HIPVs

The experiments have been well conducted and the results are clear. 

There are some improvement that should be made to the M&M, Results, and Discussion section to clarify certain passages. Some sections would need to be reviewed to improve the English language. 

Introduction:

L71 : What is an insect virus-vesicular virus ?

LL71-73 : What is a "bee-venom" coordinated control mode ?

M&M section:

2.1 Plant material: To me, it seems that single leaves are always employed in the different experiments, not whole plants. Please clarify. Also, please address the fact that healthy leaves are cut from the plant : how does this affect VOCs emissions? 

LL91-92: Suggestion : S. litura eggs and M. similis were obtained from our laboratory. M. similis were reared from S. litura larvae as the host.

2.3 Olfactory preferences of M. similis (M. similis olfactometer experiments 1–4) : consider reorganizing the sequence of information. For example, insert as the second sentence the information about the trap-balls, followed by the LED lighting and the white canvas. Make clear that the canvas is to diffuse the light. 

LL108-112: consider rewriting  = Individual air flow were connected to each odor source and converged to a central glass piece where 4 M. similis female were released. After thirty minutes following release, the location of each of the wasps was noted. Selection for an odour source was concluded when a wasp was present in one of the four ball-traps. The experiment was replicated thirty times with 4 female wasps each time.

LL119-132 : review larva (singular) versus larvae (plural)

L127: experiment was begun

L131: Replace The Petri dish and filter paper were replaced every 3 larvae.

L132 : 30 replicates

L134: arms at a 60° angle

LL135-138: why this king of information is not provided for the experiment with the parasitoid?

L138: why under dark conditions?

LL143-144: it should be made clear that these are the same treatments as for the Petri dish experiment.

2.5 Collection and analysis of volatiles via solid phase microextraction 

LL168-70: the chi-square tests apply only to the experiment with S. litura.

Results section

LL199-200: In the first Petri dish experiment, S. litura preferred healthy leaves over no leaves (blank) and preferred S. litura-infested leaves over no leaves (blank) (P < 0.001; Fig. 2a), but the selection rate tended to decrease as the infestation time increased.

LL204-206: In “Y” type olfactometer experiments, the selection rate of healthy plants and of S. litura-infested plants was significantly higher than that of no plant but an increase in the infestation time tended to decrease the selection of infested plants (P < 0.01; Fig. 2c).

Tables 1 and 2: significance of the slopes or of the regression models should be indicated.

3.6 Relationship between the release of specific volatiles by Chinese cabbage and the host searching behavior of M. similis and S. litura: results from table 2 are not presented in the text.

Discussion section

The first paragraph should start by stating the major results of the study. They appear somehow at the end of the paragraph.

These results may be made more precise.

For example : This study shows that Chinese cabbage releases volatiles that can suppress S. litura larvae from choosing leaves that are already exploited. However, when the larvae choose to feed on unattacked leaves, they face the risk of being found by the parasitoid M. similis. These patterns support the proposition that plants have evolved to manipulate the blend of volatiles that are emitted in order to maximise both direct and indirect defenses against herbivory.

The challenge in the discussion is to provide some tentative explanation why the herbivore and the parasitoid are expressing opposite attraction to different volatiles since it seems that the parasitoid is more attracted to HIVCs than to VOCs.

Reviewer 2 Report

Comments to the authors

This manuscript by Du et al., demonstrated the effects of volatiles from Spodoptera litura-infested Chinese cabbage on host location by S. litura caterpillars and its larval parasitoid wasp, Microplitis similis. The major finding of this study is that volatiles from S. litura-infested Chinese cabbage reduce preference of S. litura caterpillars for S. litura-infested Chinese cabbage, while the volatiles attract Microplitis similis wasps. In addition to that, the authors identified volatile compounds that contribute to these insects' behavior. These findings have not previously been reported. The manuscript is well written, and the results are clearly presented. I have some comments on the text, as explained below.

Major comments:
L166-L172:
Explanation on statistical tests is not clear. The authors described that "The data concerning the effects of treatments on selection of leaves, plants, and volatiles by S. litura and M. similis were compared with Chi-square test, and the volatiles released from leaves of different treatments were compared with ANOVA (P <0.05)". However, according to the legend of Fig. 1, ANOVA was used for the data concerning the effects of treatments on selection of volatiles by M. similis. In addition to that, according to Figs. 1, 3 and 4, multiple comparison were applied to these data. However, both Chi-square test and ANOVA test the overall significance of three or more independent data, while they do not tell us significant differences between each combination of data. Post-hoc test is needed.

L340-374:
It is well known that combination of HIPVs compounds are important for parasitoid/herbivore attractions. I would like the authors to mention about the possibility of synergistic effects of HIPV compounds on parasitoid/herbivore attractions.

Minor comments:
L64-65: "S. litura a serious pest," -> "S. litura is a serious pest,"
L91-92: information on original source of material insects is needed.
L113: "." -> ","
L133-: There is no information on diameter of the Y-tube olfactometer.
L208: "S. litura" should be italic font.
L442: "Satoshi, T." -> "Tatemoto, S." (Satoshi is his first name.)

Round 2

Reviewer 2 Report

The authors did not understand my comments on the statistics. Therefore, I could not find the improvement in the way of describing statistical method and figure legends. They should learn post-hoc statistical methods after ANOVA and analyze the data, then submit the manuscript again.

Round 3

Reviewer 2 Report

Statistics seems good by this correction.  (E) and (Z) should be italicized, like (E) and (Z).
